# CNN-ViT Supported Weakly-Supervised Video Segment Level Anomaly Detection

**DOI:** 10.3390/s23187734

**Published:** 2023-09-07

**Authors:** Md. Haidar Sharif, Lei Jiao, Christian W. Omlin

**Affiliations:** Department of ICT, University of Agder, 4630 Kristiansand, Norway; lei.jiao@uia.no (L.J.); christian.omlin@uia.no (C.W.O.)

**Keywords:** attention, convolutional neural network (CNN), Mahalanobis distance, multiple instance learning (MIL), vision transformer (ViT), weakly supervised video anomaly event detection

## Abstract

Video anomaly event detection (VAED) is one of the key technologies in computer vision for smart surveillance systems. With the advent of deep learning, contemporary advances in VAED have achieved substantial success. Recently, weakly supervised VAED (WVAED) has become a popular VAED technical route of research. WVAED methods do not depend on a supplementary self-supervised substitute task, yet they can assess anomaly scores straightway. However, the performance of WVAED methods depends on pretrained feature extractors. In this paper, we first address taking advantage of two pretrained feature extractors for CNN (e.g., C3D and I3D) and ViT (e.g., CLIP), for effectively extracting discerning representations. We then consider long-range and short-range temporal dependencies and put forward video snippets of interest by leveraging our proposed temporal self-attention network (TSAN). We design a multiple instance learning (MIL)-based generalized architecture named CNN-ViT-TSAN, by using CNN- and/or ViT-extracted features and TSAN to specify a series of models for the WVAED problem. Experimental results on publicly available popular crowd datasets demonstrated the effectiveness of our CNN-ViT-TSAN.

## 1. Introduction

Fully supervised, unsupervised, and weakly supervised are the three dominant paradigms in video anomaly event detection (VAED). The fully supervised paradigm mostly gives a high performance [1]. Nevertheless, frame-level normal or abnormal annotations in the training data are essential, which requires the video annotators to localize and label abnormalities in videos. As abnormalities can take place at any time, nearly all frames need to be spotted by the annotators. Unfortunately, it can be a non-automated and time-consuming process to accumulate a fully annotated large-scale dataset for supervised VAED.In the unsupervised paradigm, the models are trained on samples of normal events solely, along with a common assumption that the unseen anomaly videos will have high reconstruction errors [2,3,4]. Unluckily, the performance of unsupervised VAED is commonly inferior, due to its lack of advance understanding of anomalies, as well as its inability to capture all kinds of normality variants [5]. The weakly supervised approaches are thus considered to be the most practical paradigm, over both unsupervised and supervised paradigms, due to their competitive performance as well as annotation cost-effectiveness, by applying video-level labels to lower the cost of laborious fine-grained annotations [6,7].

Nowadays, WVAED has become an established VAED technical route of research  [6,7,8,9,10,11,12,13,14,15,16]. The WVAED problem is mainly regarded as an MIL (multiple instance learning) problem [8]. In general, WVAED models directly output anomaly scores by comparing the spatiotemporal features of normal and abnormal events through the MIL. The MIL pertains to training data organized in sets, called positive and negative bags. A video in MIL is regarded as a bag holding multiple instances, where each instance belongs to a video snippet. A negative bag contains all normal snippets, whereas a positive one contains both normal and abnormal snippets, without any temporal information about the beginning and end of abnormal events. The standard MIL assumes that all negative bags accommodate only negative snippets, and that positive bags carry no less than one positive snippet. Supervision is provided solely for complete sets, and the isolated label of the snippets contained in the bags is not provided [17]. As WVAED can understand the essential variability between normal and abnormal, its outputs are fundamentally more reliable than those of unsupervised VAED [18]. However, in WVAED, abnormal-labeled frames of the positive bag tend to be influenced by normal-labeled frames in the negative bag, while the abnormality will not certainly stand out in opposition to the normality. Subsequently, sometimes it becomes difficult to detect anomalous snippets. Many researchers (e.g., [8,9,10,19,20]) have made efforts to take this problem forward using MIL frameworks. Many of the existing approaches encode the extracted visual content by applying a backbone (e.g., C3D [21], I3D [22]), which are pretrained on action recognition tasks. However, VAD depends on discriminative representations that clearly represent the events in a scene. Thus, these existing backbones are not suitable for VAD, due to the domain gap [1]. To address this limitation, and inspired by the success of the recent vision-language works of [23,24,25], which proved the potency of feature representation learned via contrastive language-image pretraining (CLIP) [26], Joo et al. [20] employed the vision transformer (ViT) encoded visual features from CLIP [26]. However, the performance of MIL-based WVAED methods heavily depends on the pretrained feature extractors.

In this paper, we first propose utilizing pretrained feature extractors using backbones of both CNN (e.g., C3D [21], I3D [22]) and ViT (e.g., CLIP [26]) for extracting discerning representations effectively. We propose a temporal self-attention network (TSAN) to generate the reweighed attention feature by modeling the continuity between snippets of a video and selecting the top-*k* most relevant snippets. Later, the reweighed attention features are used to produce anomaly scores using a multi-layer perceptron (MLP) based score allocator. In the TSAN pipeline, we utilize the statistically most significant features as probabilities by employing a temporal scoring technique considering Mahalanobis distances instead of the mean feature magnitudes of snippets. The motivations behind the usage of the Mahalanobis metric over the mean are as follows: (i) It can correct the correlations between the different features; (ii) It automatically accounts for the scaling of the coordinate axes; (iii) It can provide curved as well as linear decision boundaries. Our ablation study showed that maximum mean of 5.34% better performance can be achieved empirically by employing the Mahalanobis metric. In addition, the TSAN also deals with an arbitrary number of abnormal snippets in an abnormal video. The top-*k* selector in the TSAN addresses k-snippets of interest in the video. We model long-range and short-range temporal dependencies and put forward the snippets of interest by supporting TSAN. In brief, we design a MIL-based generalized architecture of CNN-ViT-TSAN, as portrayed in Figure 1, to specialize five different models, namely C3D-TSAN, I3D-TSAN, CLIP-TSAN, C3D-CLIP-TSAN, and I3D-CLIP-TSAN, for WVAED problems. Each model consists of three main modules responsible for (i) Feature encoding by the CNN and/or ViT; (ii) Patterning snippet consistency in the temporal dimension using TSAN; and (iii) Identifying abnormal snippets in connection with the separation maximization supervisor (SMS), where the SMS trains the abnormal snippets to have a high value and the normal snippets to have a low value. The C3D-TSAN and I3D-TSAN models do not require ViT-based feature extraction, while the CLIP-TSAN model does not need CNN-based feature extraction. Information fusion takes place in the TSAN for C3D-CLIP-TSAN and I3D-CLIP-TSAN models only, whereas the models for C3D-TSAN, I3D-TSAN, and CLIP-TSAN skip it. Each of our proposed models is based on a distinct degree of feature extraction and usability capabilities required for crowd video anomaly detection. Consequently, in experimental setups considering UMN, UCSD-Ped1, UCSD-Ped2, ShanghaiTech, and UCF-Crime datasets, some of these models demonstrated inferior results, while others showed superior results. For example, the model I3D-CLIP-TSAN demonstrated the best results and outperformed its alternatives by extracting and using high-quality features from the available videos, as well as confirming a better normal—abnormal snippet separability.

The unique contributions and advancements that our proposed CNN-ViT-TSAN framework brings to the field of WVAED problems are recapitulated as follows:We propose five deep models for WVAED problems by designing a MIL-based generalized framework CNN-ViT-TSAN. The information fusion between CNN and ViT is a unique contribution;We propose a TSAN that helps to provide anomaly scores for video snippets in WVAED problems;We uniquely introduce the usage of the Mahalanobis metric for calculating probability scores in the TSAN;Experiments on several benchmark datasets demonstrated the superiority of our models compared with the state-of-the-art approaches.

The rest of this paper is organized as follows; Section 2 addresses the most relevant previous studies. Section 3 discusses our proposed generalized framework. Section 4 explains the experimental setup; the results obtained on public datasets; as well as a comparison, reasons for superiority, best network analysis, ablation study, and limitations of our models. Section 5 concludes the paper with a few clues for further study.

## 2. Related Work

Methods of WVAED are based on video-level labels, which always follow the MIL ranking framework [8]. Based on MIL, a method of WVAED trains a regression model to assign scores for video snippets, assuming that the maximum score of the positive bag is higher than that of the negative bag. The existing methods of WVAED can be roughly categorized into two broad kinds on the basis of the pretrained models used, namely: CNN-based and ViT-based WVAED methods, as summarized below.

### 2.1. CNN-Based WVAED Methods

Sultani et al. [8], Tian et al. [19], Zhang et al. [9], Zhong et al. [6], and Zhu et al. [11] employed CNN-based pretrained models in their experimental setups. Sultani et al. [8] also pre-collected annotated normal and abnormal video events at video-level to build their popular UCF-Crime dataset and applied it with their weakly supervised framework for detecting anomalies. In their framework, after extracting C3D features [27] for video segments, they trained a fully connected neural network by applying a ranking loss function, which computed the ranking loss between the highest scored instances in the positive bag and the negative bag. Tian et al. [19] treated C3D [27] and I3D [22] as feature extractors for their WVAED model. They claimed that the selection of the top-3 features based on their magnitude can introduce a greater partition between normal and anomalous videos, where if more than one abnormal snippet exists per anomalous video, the mean snippet feature magnitude of the anomalous videos is larger than that of normal videos.

Zhang et al. [9] trained a temporal convolution network between the preceding adjacent segment and current segment for extracting positive and negative video segment C3D features [27]. Afterwards, they trained two branches of a fully connected neural network using an inner and outer bag ranking loss, considering the highest and lowest scored segments in the positive and the negative bags. Zhong et al. [6] and Zhu et al. [11] trained both a feature encoder and classifier together. Zhong et al. [6] addressed WVAED as a supervised learning task under noise labels. However, to verify the widespread applicability of their model, they carried out extensive experiments considering a C3D [27] and a temporal segment network [28]. Zhu et al. [11] included the temporal context into their MIL ranking model by applying an attention block. They claimed that features containing motion information extracted by C3D [27] and I3D [22] performed better than features extracted from separate images using VGG16 [29] and Inception [30], regardless of the network depth and feature dimension.

### 2.2. ViT-Based WVAED Methods

ViT-based pretrained models can be categorized into single-stream or dual-stream types. The single-stream model applies a single transformer to model both image (or video) and text representations in a combined framework, whereas the dual-stream model independently encodes image (or video) and text with a decoupled encoder. Examples of ViT feature extractors include VisualBERT [31], ViLBERT [32], CLIP [26], and data efficient CLIP [33]. Recently, Joo et al. [20] proposed a CLIP-assisted [26] temporal self-attention framework for the WVAED problem. They conducted experiments on publicly available datasets to verify their end-to-end WVAED framework. Li et al. [34] suggested a transformer-based multi-instance learning network to learn video-level anomaly probability and snippet-level anomaly scores. In the inference stage, they employed the video-level anomaly probability to suppress the fluctuation of snippet-level anomaly scores. Lv et al. [35] presented an unbiased MIL scheme that learned an unbiased anomaly classifier and a tailored representation for WVAED.

In view of the existing solutions, we found that, generally, a CNN and ViT are employed separately. To take advantage of both CNN- and ViT-based pretrained models, we designed an MIL-supported generalized architecture named CNN-ViT-TSAN to specify a series of models for the WVAED problem.

## 3. Proposed Generalized Framework

Our generalized framework follows the MIL model, in which the positive bag represents an anomaly and the negative bag denotes normality. Its constituent components are discussed in the following subsections.

### 3.1. Feature Extraction

Videos in the training set are only labeled at video-level in WVAED. Assume that a set of weakly labeled training videos $W={\left\{V_{v},y_{v}\right\}}_{v=1}^{\left|W\right|}$ are available, where each video $V_{v}={\left\{Frame_{i}\right\}}_{i=1}^{N_{v}}\in R^{N_{v}\times W\times H}$ hints at a sequence of $N_{v}$ frames with *W* pixels for width and *H* pixels for height. Here, $y_{v}=\left\{0,1\right\}$ indicates the video-level label of video $V_{v}$ with respect to anomaly, i.e., it is 1 for an anomaly video that holds at least one abnormal event, otherwise 0. For a video $V_{v}={\left\{Frame_{i}\right\}}_{i=1}^{N_{v}}$, we divide it into a set of ${\left\{\gamma_{i}\right\}}_{i=1}^{\left\lfloor \frac{N_{v}}{\Delta }\right\rfloor }$ equal number of non-overlapping temporal snippets each with a length of Δ-frame.

#### 3.1.1. Feature Extraction Using a Pretrained CNN

Convolutional neural networks (CNN), as one of the most representative deep learning models, exhibit great potential in the field of image classification. CNN-based C3D (Convolutional 3D) [21] and I3D [22] are two common feature extractors. As a feature extractor, the C3D is generic, compact, simple, and efficient. Tran et al. [27] showed that C3D can model appearance and motion information simultaneously and outperformed the 2D CNN features in various video-analysis tasks. Carreira et al. [22] introduced a two-stream (i.e., RGB and Flow) Inflated 3D CNN (I3D). Ideally, feature extraction can be efficiently performed by either C3D or I3D. We considered the C3D feature of Ji et al. [21] and I3D feature of Carreira-Zisserman [22]. We computed features of *T* snippets with feature dimension ${ℵ}^{\prime }$ using both C3D and I3D separately. Let $\Phi_{v_{cnn}}^{\prime }={\left\{\phi_{i}\right\}}_{i=1}^{T_{v}}\in R^{T_{v}\times {ℵ}^{\prime }}$ be the extracted features of $V_{v}$, where $T_{v}$ belongs to the number of snippets for $V_{v}$.

For the dimensionality reduction technique, the principal component analysis (PCA) works under the assumption that the data follow a normal distribution. For this reason, they may be very sensitive to the variance of the variables. In addition, as the extracted data are not normalized, the reduced dimensions using PCA or other similar techniques would give erroneous results. However, the low-variance-filter is an advantageous dimensionality reduction algorithm often used in machine learning on numerical data. Instead of using PCA, we apply the low-variance-filter algorithm to reduce the dimensionality of the extracted data. Upon dimensionality reduction, $\Phi_{v_{cnn}}^{\prime }\in R^{T_{v}\times {ℵ}^{\prime }}$ can obtain the shape of $\Phi_{v_{cnn}}\in R^{T\times ℵ}$, i.e.,  *ℵ*-dimensional feature of the *T* snippets.

#### 3.1.2. Feature Extraction Using a Pretrained ViT

Vision-language pretrained models extract the relationships between objects/actions in a video and objects/actions in text using vision transformers (ViTs). Based on the suitability of applications, various kinds of ViTs exist, e.g., VisualBERT [31], ViLBERT [32], CLIP [26], and data efficient CLIP [33]. In general, CLIP [26] is a multi-modal vision and language model, which utilizes a ViT as a backbone for visual features. We assume that the middle frame $d_{j}=\left\lceil \frac{\Delta }{2}\right\rceil$ represents the snippet $\gamma_{j}$, instead of considering all frames in a snippet $\gamma_{j}$. Following Joo et al. [20], we apply CLIP [26] to the $d_{j}$ of the snippet $\gamma_{j}$ to represent its feature as $\phi_{j}\in R^{ℵ}$ with feature dimensions *ℵ*, and then $V_{v}$ can be constituted as a set of video feature vectors $\Phi_{v_{vit}}={\left\{\phi_{j}\right\}}_{j=1}^{T_{v}}\in R^{T\times ℵ}$.

### 3.2. Temporal Self-Attention Network (TSAN)

Figure 1 visualizes our proposed TSAN mechanism, which models the snippet coherency and selects the top-*k* most significant snippets. It maximizes the attention on a subset of features, while it minimizes attention on noise. The pipeline of TSAN consists of four components namely: (i) a temporal scoring module, (ii) top-*k* selecting module, (iii) multiplying-averaging module, and (iv) information fusion module.

#### 3.2.1. Temporal Scoring Module

The temporal scoring technique utilizes the statistically most significant features as probabilities, considering Mahalanobis distances instead of the mean feature magnitudes of the snippets. The mathematical exposition is given in Algorithm 1. The scores of $P_{\mathrm{score}}\in$$R^{T\times 1}$ are employed to estimate anomaly attention features, upon extracting the *k* most significant snippets from the video using Algorithm 2. Concisely, each of $\Phi_{v_{cnn}}\in R^{T\times ℵ}$ and $\Phi_{v_{vit}}\in R^{T\times ℵ}$ can be converted into a probability score vector $P_{\mathrm{score}}\in R^{T\times 1}$ using Algorithm 1, where each score represents a snippet. The scores of $P_{\mathrm{score}}\in$$R^{T\times 1}$ are fed to the top-*k* selector module for further processing. The model CLIP-TSAN does not expect the $\Phi_{v_{cnn}}\in R^{T\times ℵ}$ to be processed using Algorithm 1 to obtain $P_{\mathrm{score}}\in R^{T\times 1}$. In this case, the final output $\bar{\Phi_{v_{cnn}}}\in R^{T\times ℵ}$ of the multiplying-averaging module has no active function in the information fusion module. Thus, solely $\Phi_{v_{vit}}\in R^{T\times ℵ}$ is processed using Algorithm 1 to obtain $P_{\mathrm{score}}\in R^{T\times 1}$ for feeding to the top-*k* selecting module. Conversely, the model of C3D-TSAN does not look for the $\Phi_{v_{vit}}\in R^{T\times ℵ}$ to be processed using Algorithm 1 to obtain $P_{\mathrm{score}}\in R^{T\times 1}$. In this instance, the final output $\bar{\Phi_{v_{vit}}}\in R^{T\times ℵ}$ of the multiplying-averaging module has no operational function in the information fusion module. Consequently, only $\Phi_{v_{cnn}}\in R^{T\times ℵ}$ is processed considering Algorithm 1 to obtain $P_{\mathrm{score}}\in R^{T\times 1}$. Likewise, the model I3D-TSAN does not expect the scores of $P_{\mathrm{score}}\in$$R^{T\times 1}$ obtained from $\Phi_{v_{vit}}\in R^{T\times ℵ}$. However, the models of C3D-CLIP-TSAN and I3D-CLIP-TSAN need the scores of $P_{\mathrm{score}}\in$$R^{T\times 1}$ obtained from both $\Phi_{v_{cnn}}\in R^{T\times ℵ}$ and $\Phi_{v_{vit}}\in R^{T\times ℵ}$. They use Algorithm 1 to obtain $P_{\mathrm{score}}\in R^{T\times 1}$ in a sequential manner, such as in CLIP-TSAN, C3D-TSAN, and/or I3D-TSAN. In the case of either C3D-CLIP-TSAN or I3D-CLIP-TSAN, the final outputs of $\bar{\Phi_{v_{cnn}}}\in R^{T\times ℵ}$ and $\bar{\Phi_{v_{vit}}}\in R^{T\times ℵ}$ from the multiplying-averaging module are stored in the information fusion module for element-wise addition.
**Algorithm 1:** Calculation of the probability scores $P_{\mathrm{score}}$ considering Mahalanobis distances
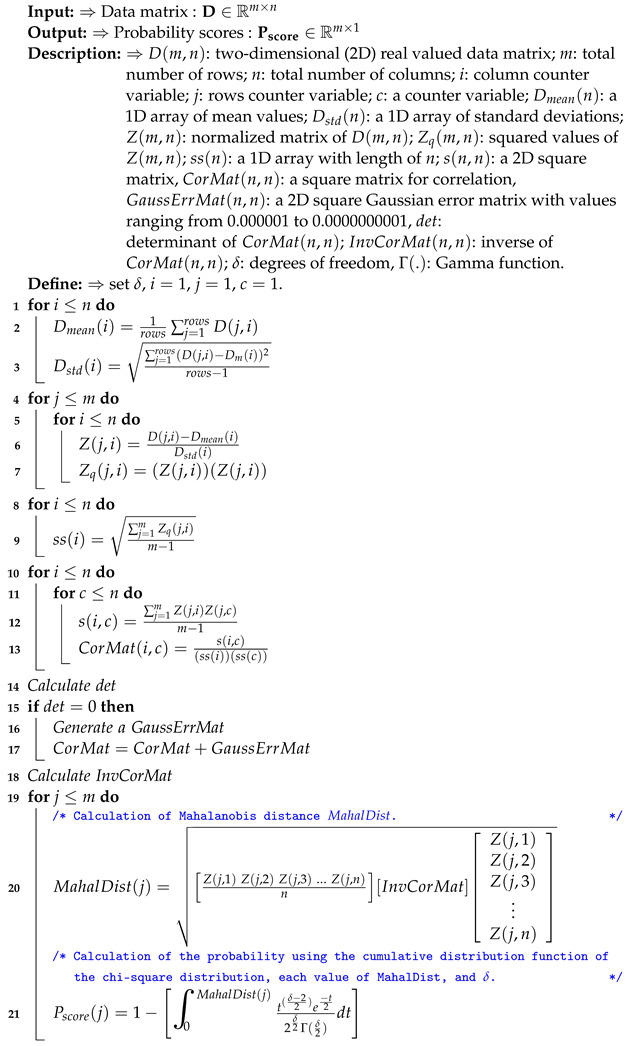

**Algorithm 2:** Processing of the probability scores $P_{\mathrm{score}}$ in the TSAN
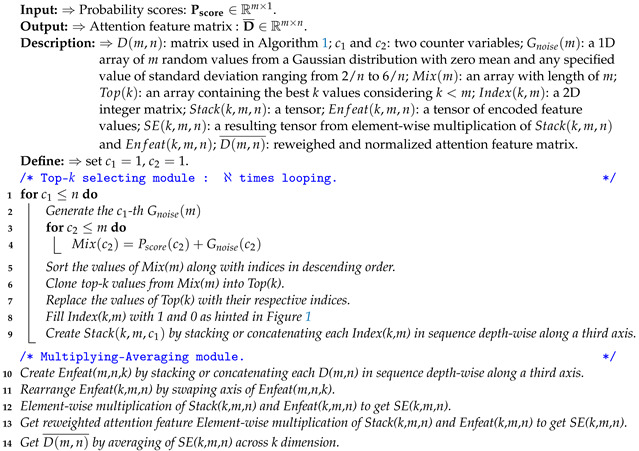


#### 3.2.2. Top-*k* Selecting Module

This module extracts the k<T most interesting snippets from a video. The value of *k* is determined using Equations (Equation 1) and (Equation 2) as
(1)$$k=\left\lfloor \frac{\mu \;\;\log_{2}\left(T\right)}{\log_{e}\left(2\right)}\;\;\sin\left(\frac{H}{W}\right)\right\rfloor ,$$
(2)$$\mu =\left\lceil \frac{1}{\sqrt{2}}+\frac{\log_{2}\left(T\right)}{2}\right\rceil .$$

A specific Gaussian noise score vector G∈$R^{T\times 1}$ is generated to apply to the scores of $P_{\mathrm{score}}\in$$R^{T\times 1}$ for producing Gaussian-perturbed scores of $G_{\mathrm{per}}\in R^{T\times 1}$ using Equation (Equation 3) as
(3)$$G_{\mathrm{per}}\in R^{T\times 1}=P_{\mathrm{score}}\in R^{T\times 1}\;\;\;⨁\;\;\;\;\;\;G\in R^{T\times 1},$$
where ⨁ belongs to an element-wise addition. The values of $G_{\mathrm{per}}\in R^{T\times 1}$ are sorted along with the indices in descending order. Replacement of the *k*-best values with their respective indices is achieved as shown in Figure 1. For example, if the 1st *k*-best value has the index of T−2, then this 1st *k*-best value will be replaced by T−2. Afterwards, the value of its corresponding 2D matrix’s ${\left(T-2\right)}^{th}$ column of the 0th row will be filled with 1, but all other columns of the 0th row will be filled with 0. Similarly, if the 2nd *k*-best value has an index of 0, then this 2nd *k*-best value will be replaced by 0. The value of its corresponding 2D matrix’s 0th column for the 1st row will be filled with 1 but all other columns of the 1st row will be filled with 0, and so on. However, the aforementioned procedure is repeated *ℵ* times and the results are stacked or concatenated to obtain a 3D tensor, which is fed to the multiplying-averaging module. Line 1 to Line 9 of Algorithm 2 represent the top-*k* selecting module.

#### 3.2.3. Multiplying-Averaging Module

Taking into account $\Phi_{v_{cnn}}\in R^{T\times ℵ}$ or $\Phi_{v_{vit}}\in R^{T\times ℵ}$, a 3D tensor is created by cloning *k* times of T×ℵ. The tensor is reshaped to perform an element-wise multiplication with the output of the 3D tensor of the top-*k* selecting module. The final product is converted from a 3D tensor to 2D matrix using an averaging technique. Line 10 to Line 14 of Algorithm 2 illustrate the multiplying-averaging module. The output of Algorithm 2 is a reweighed and normalized anomaly attention feature matrix.

#### 3.2.4. Information Fusion Module

This module holds any output of the multiplying-averaging module (i.e., Algorithm 2) for a video $V_{v}$. Explicitly, the reweighed and normalized anomaly attention feature matrices of $\bar{\Phi_{v_{cnn}}}\in R^{T\times ℵ}$ and $\bar{\Phi_{v_{vit}}}\in R^{T\times ℵ}$ are stored in two memory locations. Based on our five different modeling options, the information of $\bar{\Phi_{v_{cnn}}}\in R^{T\times ℵ}$ and $\bar{\Phi_{v_{vit}}}\in R^{T\times ℵ}$ either can or cannot be fused. In the case of the C3D-TSAN, I3D-TSAN, and CLIP-TSAN models, the information fusion of $\bar{\Phi_{v_{cnn}}}\in R^{T\times ℵ}$ and $\bar{\Phi_{v_{vit}}}\in R^{T\times ℵ}$ is not required.Conversely, in the case of the C3D-CLIP-TSAN and I3D-CLIP-TSAN models, the information fusion of $\bar{\Phi_{v_{cnn}}}\in R^{T\times ℵ}$ and $\bar{\Phi_{v_{vit}}}\in R^{T\times ℵ}$ takes place by considering the mode of element-wise addition.

### 3.3. Training Phase

In the MIL framework, accurate temporal locations of abnormal events in videos are unspecified. Instead, only video-level labels specifying the existence of an abnormal event in the whole video is needed. A video is called a bag. It is labeled as a positive bag if it holds a minimum of one snippet of an abnormal event, otherwise it is labeled as a negative bag. In the negative bag, none of the snippets contain an abnormal event. The concept is that the anomalous snippets have higher anomaly scores than the normal snippets.

We normalized the video feature length for training. The training of a mini-batch may face problems due to the difference in video embedding feature length *T* between samples in the mini-batch. Suppose that the video feature vectors of videos $V_{v-2}$ and $V_{v-1}$ are $\Phi_{v-2}={\left\{\phi_{i}\right\}}_{i=1}^{T_{k-2}}$ and $\Phi_{k-1}={\left\{\phi_{j}\right\}}_{j=1}^{T_{k-1}}$, respectively, where $T_{v-2}\neq T_{v-1}$. It is difficult to train the features in the batches due to $T_{v-2}\neq T_{v-1}$, i.e., the lack of a uniform shape in temporal dimension. Explicitly, it is important to reshape $T_{v-2}$ and $T_{v-1}$ into the same size of *T*. To handle an arbitrary length of videos in the training phase only, we follow the same normalization technique as Sultani et al. [8] and Joo et al. [20]. As the testing videos are assessed individually, we assume that it is not required to send the features through the normalization process in the testing phase.

Assume that an input mini-batch of 2Ψ videos ${\left\{V_{v}\right\}}_{v=1}^{2\Psi }$ is available, as shown in Figure 1, where none of the first half ${\left\{V_{v}\right\}}_{v=1}^{\Psi }$ contains an anomaly snippet and at least one (or more) of the snippets contains an anomaly in the second half ${\left\{V_{v}\right\}}_{v=\Psi }^{2\Psi }$. Let $\Upsilon \in R^{2\Psi \times T\times ℵ}$ indicate the extracting features upon using pretrained feature extractors in Section 3.1. During training, the first half of the mini-batch $\Upsilon_{n}\in R^{\Psi \times T\times ℵ}$, which has none of the snippets containing an anomaly feature, is loaded with a set of negative bags, while the second half $\Upsilon_{a}\in R^{\Psi \times T\times ℵ}$, which has at least one of the snippets containing an anomaly feature, is loaded with a set of positive bags in order within the mini-batch. Subsequently, both $\Upsilon_{n}\in R^{\Psi \times T\times ℵ}$ and $\Upsilon_{a}\in R^{\Psi \times T\times ℵ}$ go through the stage of TSAN. The output of TSAN is a set of reweighed normal attention features $\bar{\Phi_{n}}$, as well as a set of reweighed anomaly attention features $\bar{\Phi_{a}}$. Then the reweighed attention features undergo the snippet association network, which consists of a pyramid of dilated convolutions [36] and non-local block [37], to determine the long-term and short-term association between snippets, in accordance with the reweighed magnitudes of $\bar{\Phi_{n}}$ and $\bar{\Phi_{a}}$. The output of the snippet association network is the final attention features $\bar{\Phi_{f}}\in R^{2\Psi \times T\times ℵ}$, which are then passed to a layered MLP-based score converter that converts the feature vectors into a set of 2ΨT anomaly scores. This set of scores is used for computing the score-based binary cross-entropy loss.

Let $p_{i}$ be the anomaly score of the $i^{th}$ snippet. Given the snippet-wise anomaly scores ${\left\{{\tilde{z}}_{i}\right\}}_{i=1}^{2\Psi T}$, the cross-entropy loss over the top-*k* snippets can be calculated using Equation (Equation 4) as
(4)$$Cross\;entropy\;loss=-\frac{1}{\left|\varpi \right|}\sum_{i\in \varpi }^{}\left[{\tilde{z}}_{i}\;\log\left(p_{i}\right)+\left(1-{\tilde{z}}_{i}\right)\;\log\left(1-p_{i}\right)\right],$$
where ϖ belongs to the set of top-*k* snippets.

### 3.4. Separation Maximization Supervisor (SMS) Learning

We employ a SMS, denoted by $\xi_{\tau ,\mu }$, to maximize the separation between the top snippets of the positive and negative bags. Each attention feature of $\bar{\Phi_{f}}\in R^{2\Psi \times T\times ℵ}$, irrespective of normal and abnormal bags, undergoes SMS. First, $\bar{\Phi_{f}}\in R^{2\Psi \times T\times ℵ}$ is rearranged to make it suitable for SMS processing by specifying $\bar{\Phi_{f+}}\in R^{\Psi \times T\times ℵ}$ and $\bar{\Phi_{f-}}\in R^{\Psi \times T\times ℵ}$ as anomaly and normal attention features, respectively. Then, we select the top-μ snippets from $\bar{\Phi_{f+}}\in R^{\Psi \times T\times ℵ}$ and $\bar{\Phi_{f-}}\in R^{\Psi \times T\times ℵ}$ using feature magnitude. This produces $\bar{\Phi_{f{+}_{\mu }}}\in R^{\Psi \times \mu \times ℵ}$ and $\bar{\Phi_{f{-}_{\mu }}}\in R^{\Psi \times \mu \times ℵ}$ as subsets of $\bar{\Phi_{f+}}\in R^{\Psi \times T\times ℵ}$ and $\bar{\Phi_{f-}}\in R^{\Psi \times T\times ℵ}$, respectively. Then, both $\bar{\Phi_{f{+}_{\mu }}}\in R^{\Psi \times \mu \times ℵ}$ and $\bar{\Phi_{f{-}_{\mu }}}\in R^{\Psi \times \mu \times ℵ}$ are averaged out across the top-μ snippets to produce $\beta_{\tau ,\mu }\left(\bar{\Phi_{f+}}\right)\in R^{\Psi \times ℵ}$ and $\beta_{\tau ,\mu }\left(\bar{\Phi_{f-}}\right)\in R^{\Psi \times ℵ}$ to represent the Ψ-anomaly and Ψ-normal bags, each with a feature vector of length *ℵ*, respectively. Both $\beta_{\tau ,\mu }\left(\bar{\Phi_{f+}}\right)$ and $\beta_{\tau ,\mu }\left(\bar{\Phi_{f-}}\right)$ depend on the parameters of τ, as well as μ. The τ indicates the dependency on the snippet association network, whereas μ points to the selection of the top-μ snippets with the largest temporal feature magnitude. The separability is computed using Equation (Equation 5) as
(5)$$\xi_{\tau ,\mu }\left(\bar{\Phi_{f+}},\bar{\Phi_{f-}}\right)=∥\beta_{\tau ,\mu }\left(\bar{\Phi_{f+}}\right)∥-∥\beta_{\tau ,\mu }\left(\bar{\Phi_{f-}}\right)∥.$$ Equation (Equation 5) maximizes the separability of the top-μ feature snippets from each positive and negative bag by leveraging the theorem of Tian et al. [19].

### 3.5. Loss Optimization

The feature vectors of $\xi_{\tau ,\mu }\left(\bar{\Phi_{f+}},\bar{\Phi_{f-}}\right)$ can be averaged across *ℵ* dimensions to obtain Ψ numerical values of separability for the mini-batch of 2Ψ training videos. These numerical values are averaged out and then used as a portion of the optimized loss. Basically, this portion of loss, as well as the score-based binary cross-entropy loss computed using Equation (Equation 4) for 2ΨT anomaly scores are applied to optimize the total loss of the model.

### 3.6. Testing Phase

During testing, we assumed that the extracted video feature vectors need not move through the feature length normalization process to be reshaped to the common size of *T*, as the testing videos were assessed independently. The extracted video feature vectors went through TSAN processing to generate the reweighed attention features. They were then passed into the snippet association network, followed by the MLP-based feature vector to anomaly score converter, to obtain a set of scores. Each of these scores portrays the anomaly probability of the snippet at the associated index and conveys a numerical value between 0 and 1. Each score is repeated Δ times to replicate a vector with the usual frame length of the video. It also preserves the original order of the video and utilizes an evaluation against the ground truth labels.

## 4. Experimental Setup and Results

### 4.1. Used Datasets

We evaluated our models on the following benchmark datasets:

#### 4.1.1. UMN Dataset

This dataset [38] comprises five dissimilar staged videos, where people walk around and eventually start running in distinct directions. The abnormal events are characterized by running episodes.

#### 4.1.2. UCSD-Peds Dataset

This is a small-scale dataset, which consists of two sub-datasets; namely, UCSD-Ped1 with 70 videos and UCSD-Ped2 with 28 videos. These videos were captured at one location. The anomalies in the videos are straightforward, including people walking across a walkway, non-pedestrian entities (e.g., a skater, biker, and wheelchair) in the walkways. The default training set of UCSD-Peds does not contain anomaly videos. Preceding works [6,19,39] reorganized and utilized the dataset for weakly supervised anomaly detection by randomly selecting six anomaly videos and four normal videos for the training set, with the remainder as a testing set.

#### 4.1.3. ShanghaiTech Dataset

This is a medium-scale dataset, which contains 317,398 frames of video clips, encompassing scenes of multiple areas of the ShanghaiTech Campus. It has 13 different background scenes, with 307 normal videos and 130 anomaly videos. The earliest dataset [40] is a common benchmark used to detect video anomaly events. The training set contains only normal videos. The testing set contains both normal and anomalous videos. Zhong et al. [6] rearranged the dataset by choosing a subset of anomalous testing videos as training data, to build a weakly supervised training set, such that both the training and testing sets covered all 13 background scenes. We used exactly the same procedure as Zhong et al. [6] to convert the ShanghaiTech dataset to the weakly supervised setting.

#### 4.1.4. UCF-Crime Dataset

This is a large-scale anomaly detection dataset [8], which contains 1900 untrimmed videos with a total duration of 128 h from real-world street and indoor surveillance cameras. It covers 13 real-world anomalies including abuse, arrest, arson, assault, accident, burglary, explosion, fighting, robbery, shooting, stealing, shoplifting, and vandalism. Unlike the static background in ShanghaiTech [40], UCF-Crime [8] consists of complicated and diverse backgrounds. The dataset contains 1610 (i.e., 800/810:normal/anomalous) training videos with video-level labels and 290 (i.e., 150/140: normal/anomalous) testing videos with frame-level labels.

### 4.2. Implementation Details

Following Sultani et al. [8], Tian et al. [19], and Joo et al. [20], each video was divided into 32 snippets (i.e., T=32) with the snippet length set to Δ=16 frames and μ=4 by following Equation (Equation 2). Referring to Equation (Equation 1), the values of *k* were 19, 17, 24, and 19 for UMN, Peds, ShanghaiTech, and UCF-Crime, respectively. Each mini-batch consisted of samples from 32 randomly selected normal and abnormal videos. We employed C3D [21], I3D [22], and CLIP [26] for feature extraction. The *ℵ* was set as 512 for all experiments with ${ℵ}^{\prime }>ℵ$. The thresholds of the low-variance-filter were 0.00723, 0.00835, 0.00875, and 0.00911 for the datasets of UMN, Peds, ShanghaiTech, and UCF-Crime, respectively. The three-layered MLP of 512, 256, and 1 units with its hidden layer was followed by a ReLU activation function, and its final layer was followed by a sigmoid function, to produce a value between 0 and 1. Our model was trained in an end-to-end manner and implemented using PyTorch [41]. We used the Adam optimizer [42] with a weight decay of 0.0005 and a batch size of 32 for 50 epochs. The learning rate was set to 0.001 for all datasets. We employed an Intel${}^{®}$ Core^TM^ i7-7800X CPU @3.50 GHz, along with an NVIDIA graphics card GeForce GTX 1080 for both training and evaluation of the model. We also adopted OpenAI, Google Colab, and Google Drive for feature extraction. We used the area under the receiver operating characteristic (ROC) curve (AUC) to evaluate the overall model performance. The 0≤AUC≤1 is one of the most frequently used metrics for evaluating various flows and events in crowd videos [8,43,44]. The predictions of a model were 100% wrong or correct if AUC=0 or AUC=1, respectively. Intuitively, a larger AUC implies a larger margin between the normal and abnormal snippet predictions, thus resulting in a better anomaly classifier. The sensitivity, recall, hit rate, and true positive rate (TPR) can be formulated using Equation (Equation 6) as
(6)$$TPR=\frac{t_{p}}{t_{p}+f_{n}},$$
where $t_{p}$ and $f_{n}$ specify the number of true positive frames and the number of false negative frames, respectively. The fall-out or false positive rate (FPR) can be formulated using Equation (Equation 7) as
(7)$$FPR=1-\frac{t_{n}}{t_{n}+f_{p}}=\frac{f_{p}}{f_{p}+t_{n}},$$
where $f_{p}$ and $t_{n}$ indicate the number of false positive frames and the number of true negative frames, respectively. The ROC curve is a two-dimensional graphical visualization, in which the FPR is plotted on the X-axis and the TPR is plotted on the Y-axis (e.g., right side subgraphs of Figure 2). The values of AUC are calculated as the areas below the ROC curves (e.g., the yellow colored regions of Figure 2). Mathematically, the value of AUC can be calculated using the trapezoidal numerical integration method [45].

### 4.3. Results on Various Datasets

As real-world abnormal events are miscellaneous and hard to predict, to demonstrate the applicability of our generalized framework to multiple environments, we ran experiments on frequently used VAED evaluation datasets, e.g., UMN, UCSD-Ped1, UCSD-Ped2, ShanghaiTech, and UCF-Crime. Figure 2 visualizes the sample testing results of I3D-CLIP-TSAN (Ours) with videos from the UMN, UCSD-Ped1, UCSD-Ped2, ShanghaiTech, and UCF-Crime datasets, including abnormal events with sudden running of people, vehicles passing between bidirectional flows of people, bicycle riders in a pedestrian zone, bicycles crossing, and the action of taking something from a person forcefully as well as unlawfully, respectively. The obtained frame-level AUC scores of the sample testing videos in Figure 2 were 0.991, 0.943, 0.986, 0.989, and 0.912, consecutively. Although UMN, UCSD-Ped1, and UCSD-Ped2 are popular benchmarks for video anomaly detection, they are small in terms of number of videos and the duration of the video. Alterations in the anomalies are also very narrow. Furthermore, some abnormalities are not practical or sometimes the spatial annotation is not very clear. For these reasons, few authors have conducted experiments with these datasets explicitly. However, we considered all these datasets, to show the generalizability of our models. From Figure 2, it is noticeable that I3D-CLIP-TSAN (ours) was suitable for detecting various anomaly events, ranging from simple datasets (e.g., UMN, UCSD-Ped1, and UCSD-Ped2) to large-scale datasets (e.g., ShanghaiTech and UCF-Crime).

### 4.4. Performance Comparison

Assume that $AUC_{o}$ denotes the AUC computed on the overall testing videos in a dataset. Table 1 compares the frame-level $AUC_{o}$ performance scores of our models for the UCSD-Ped2, ShanghaiTech, and UCF-Crime datasets, along with state-of-the-art methods. It seems that our proposed models could be generalized for detecting various abnormal events from those datasets. In general, both ShanghaiTech and UCF-Crime would be called wide-scale anomaly detection datasets. All authors in Table 1 considered the ShanghaiTech and UCF-Crime datasets for conducting their experiments.

The reported results in Table 1 indicate that the improvements in performance by our proposed methods on the ShanghaiTech and UCF-Crime datasets were more remarkable than those for the UCSD-Ped2 dataset. However, for a coherent and intelligible comparaison of the performance of the various methods, we performed a non-parametric statistical investigation based on the results presented in Table 1, considering two categories: the first category consisted of ShanghaiTech and UCF-Crime datasets only, while the second category consideredthe UCSD-Ped2, ShanghaiTech, and UCF-Crime datasets.

Figure 3 depicts the Nemenyi [64] post hoc critical distance diagram at a level of significance of α=0.05, considering $1-AUC_{o}$ scores in Table 1 for the first category with the existing models of Sultani et al. (2018) [8], Zhong et al. (2019) [6], Zhang et al. (2019) [9], Zaheer et al. (2020) [46], Zaheer et al. (2020) [7], Wan et al. (2020) [47], Purwanto et al. (2021) [13], Tian et al. (2021) [19], Majhi et al. (2021) [48], Wu et al. (2021) [49], Yu et al. (2021) [50], Lv et al. (2021) [12], Feng et al. (2021) [51], Zaheer et al. (2022) [3], Zaheer et al. (2022) [52], Joo et al. (2022) [20], Cao et al. (2022) [53], Li et al. (2022) [34], Cao et al. (2022) [54], Tan et al. (2022) [55], Li et al. (2022) [34], Yi et al. (2022) [56], Yu et al. (2022) [57], Gong et al. (2022) [58], Majhi et al. (2023) [59], Park et al. (2023) [60], Pu et al. (2023) [61], Lv et al. (2023) [35], Sun et al. (2023) [62], and Wang et al. (2023) [63]. If the distance between the two models is less than the Nemenyi [64] post hoc critical distance at a certain *p*-value (e.g., 0.05), there is no statistically significant difference between them. Explicitly, two models are considered significantly different if their performance variation is greater than the Nemenyi [64] post hoc critical distance. To this end, from Figure 3, it is noticeable that at α=0.05, none of the model pairs are statistically significant, as the heavy red line of length 51.7871 (which is called the Nemenyi [64] post hoc critical distance) is greater than the heavy pink line. For example, the distance between the hypothesis of I3D-CLIP-TSAN (ours) vs. Sultani et al. 2018 (C3D) [8] is |44−1|=43 (heavy pink line), which is less than 51.7871 at α=0.05 (i.e., 95% confidence limit). In other words, their distance difference was lacking by a numerical value of |51.7871−43|=8.7871. Consequently, I3D-CLIP-TSAN (ours) and Sultani et al. 2018 (C3D) [8] were not statistically significant. Similarly, the hypothesis on the difference by Joo et al. 2022 (CLIP) [20] vs. Sultani et al. 2018 (C3D) [8] was not statistically significant, as their distance difference was lacking by a numerical value of |51.7871+3−44|=10.7871. However, the model I3D-CLIP-TSAN (ours) was 1−8.7871/10.7871=18.54%, more statistically significant than that of Joo et al. 2022 (CLIP) [20].This implies that I3D-CLIP-TSAN (ours) was slightly better generalized for divergent anomaly event detection from videos from the ShanghaiTech and UCF-Crime datasets than any other model in Table 1.

Figure 4 shows a Nemenyi [64] post hoc critical distance diagram at the level of significance α=0.10 considering the $1-AUC_{o}$ scores in Table 1 for the second category with the existing models of Zhong et al. (2019) [6], Zaheer et al. (2020) [46], Tian et al. (2021) [19], and Zaheer et al. (2022) [3]. Few models fell into this category, due to the avoidance of the UCSD-Ped2 dataset by many authors. However, from Figure 4, it is noticeable that the result of the difference of I3D-CLIP-TSAN (Ours) vs. Zaheer et al. 2022 (C3D) is statistically significant, as their distance difference (i.e., |9.6667−1.3333|=8.3334) was greater than 7.2184 at a 90% confidence limit. Similarly, the results for the differences of I3D-CLIP-TSAN (Ours) vs. Zhong et al. 2019 (TSN) and C3D-CLIP-TSAN (Ours) vs. Zaheer et al. 2022 (C3D) were statistically significant. However, other results for the differences of this category were not statistically significant, as their distance differences were less than 7.2184.

In summery, some of our proposed methods demonstrated their superiority among many existing state-of-the-art methods, as indicated in Table 1. Notably, the aforementioned statistical analysis shows that the method I3D-CLIP-TSAN (ours) took the top place in the rankings of each category. This implies that I3D-CLIP-TSAN (ours) has the ability to utilize good features from the pretrained CNN-ViT feature extractors considering the available videos and confirmed the high disconnectedness between the standard and abnormal snippets for VAED.

### 4.5. Reasons for Superiority

#### 4.5.1. Advantage of Information Fusion

In TSAN, both CNN- and ViT-related processing can produce their own reweighed attention features (e.g., $\Phi_{v_{cnn}}\in R^{T\times ℵ}$ and $\Phi_{v_{vit}}\in R^{T\times ℵ}$), which can be directly used by C3D-TSAN, I3D-TSAN, and CLIP-TSAN models, as the features can individually provide necessary (but possibly not sufficient) information for producing the anomaly scores used for anomaly detection. However, the information fusion (e.g., $\Phi_{v_{fusion}}\in R^{T\times ℵ}=$$\Phi_{v_{cnn}}\in R^{T\times ℵ}+\Phi_{v_{vit}}\in R^{T\times ℵ}$) of these two atypical backbones can augment the quality of feature representation. Both the C3D-CLIP-TSAN and I3D-CLIP-TSAN models applied $\Phi_{v_{fusion}}\in R^{T\times ℵ}$ and achieved superior performance, as compared to the other models. For example, from Table 1, using the UCF-Crime dataset, the model I3D-CLIP-TSAN (ours) achieved a 1−0.8897/0.8650≈3% and 1−0.8897/0.8763≈2% better performance with respect to I3D-TSAN (ours) and CLIP-TSAN (ours), respectively. Clearly, the performance gains of 3% and 2% for I3D-CLIP-TSAN (ours) were the contribution of the information fusion in the TSAN.

#### 4.5.2. Better Information Gains with the Mahalanobis Metric

Tian et al. [19] assumed that the mean feature magnitude of abnormal snippets is larger than that of the normal snippets. However, we applied the measure of Mahalanobis distances, which is much larger and more accurate than that of the mean feature magnitudes. We provide a simple example using the UMN dataset [38].

Usually, any video from the UMN dataset [38] starts with a normal event and ends with an abnormal event. Assume that we obtained the spatiotemporal information of each frame *f* (where f∈{1,2,…,900}) in a video (e.g., third video) from the UMN dataset [38] using an existing optical-flow method. For any *f*, irrespective of normal or abnormal events, we consider the spatiotemporal information of five features that are observed in time and put in the form of a matrix M∈$R^{n\times 5}$, as follows: (8)$$M(u)(v)=\left[\begin{array}{ccccc} x(1)(1) & x(1)(2) & x(1)(3) & x(1)(4) & x(1)(5)\\ . & . & . & . & .\\ x(i)(1) & x(i)(2) & x(i)(3) & x(i)(4) & x(i)(5)\\ . & . & . & . & .\\ x(n)(1) & x(n)(2) & x(n)(3) & x(n)(4) & x(n)(5) \end{array}\right],$$
where u∈{1,2,⋯,n}; i∈u; v∈{1,2,3,4,5}; x(i)(1) ↦ *x*-coordinate of *i*; x(i)(2) ↦ *y*-coordinate of *i*; x(i)(3) ↦ *x*-velocity of *i*; x(i)(4) ↦ *y*-velocity of *i*; and x(i)(5)↦ resulting motion direction of *i*.

We calculate the sum of the mean feature magnitudes of *f* denoted as $S_{mean}\left(f\right)$ and the sum of Mahalanobis distances (considering Algorithm 1) denoted as $S_{Mahal}\left(f\right)$ using Equations (Equation 9) and (Equation 10), respectively:(9)$$S_{mean}\left(f\right)=\sum_{i=1}^{5}\left(\frac{1}{n}\sum_{i=1}^{n}M\left(i\right)\left(j\right)\right),$$
(10)$$S_{Mahal}\left(f\right)=\sum_{i=1}^{n}MahalDist\left(i\right).$$

Figure 5 shows a numerical comparison of the sum of mean feature magnitudes and the sum of Mahalanobis distances for a video from the UMN dataset [38]. It is noticeable that the normal and abnormal frames cannot be marked using mean feature magnitudes, whereas the Mahalanobis distances can somewhat find them.Thus, the Mahalanobis distance is more accurate for the ground truth than the mean feature magnitudes. We estimated the probabilities of $S_{mean}\left(f\right)$ and $S_{Mahal}\left(f\right)$ using Equations (Equation 11) and (12), respectively, as
(11)$$\begin{array}{cc} P_{mean}\left(f\right) & =4\;e^{-\sqrt{\left(\frac{S_{mean}\left(f\right)}{65}\right)}}, \end{array}$$
(12)$$\begin{array}{cc} P_{Mahal}\left(f\right) & =4\;e^{-\sqrt{\left(\frac{S_{Mahal}\left(f\right)}{65}\right)}}. \end{array}$$

In machine learning, the information gain is defined as the amount of information gained for a random variable or a signal from observing another random variable. For such a measure, Kullback—Leibler divergence $D_{\mathrm{KL}}\left(P_{Mahal}\left(:\right)\|P_{mean}\left(:\right)\right)$ [65] can be applied, where the distributions of $P_{Mahal}\left(:\right)$ and $P_{mean}\left(:\right)$ include probability values of 900 frames. Equation (Equation 13) can be called the information gain achieved, if $P_{Mahal}\left(:\right)$ is employed as an alternative to $P_{mean}\left(:\right)$. If $P_{Mahal}\left(:\right)$ and $P_{mean}\left(:\right)$ perfectly match, then $D_{\mathrm{KL}}\left(P_{Mahal}\left(:\right)\|P_{mean}\left(:\right)\right)=0$, or else it can take values between 0 and *∞*.
(13)$$\begin{array}{c} D_{\mathrm{KL}}\left(P_{Mahal}\left(:\right)\|P_{mean}\left(:\right)\right)=\sum_{f=1}^{900}\left(\left(P_{Mahal}\left(f\right)\right)\left(\log\left(\frac{P_{Mahal}\left(f\right)}{P_{mean}\left(f\right)}\right)\right)-P_{Mahal}\left(f\right)+P_{mean}\left(f\right)\right). \end{array}$$

The calculated score of 118.41 in Equation (Equation 13) quantifies how much the probability distribution of $P_{Mahal}\left(:\right)$ differs from the $P_{mean}\left(:\right)$ probability distribution on identical grounds. Explicitly, the information gain achieved by $P_{Mahal}\left(:\right)$ with respect to $P_{mean}\left(:\right)$ was about 118. To keep pace with ground truth, the sum of the mean feature magnitudes for an abnormal event should be either greater or lesser than that of a normal event, but Figure 5a does not reflect this. On the other hand, to keep pace with the ground truth, the sum of Mahalanobis distances for an abnormal event should be either greater or lesser than that of a normal event, and Figure 5b reflects this. As Figure 5 shows that the measure of Mahalanobis distance is closer to the ground truth, the measure of Mahalanobis distance is more accurate than that of the mean feature magnitudes. The practical results for different datasets on identical grounds also reflected this proposition.

### 4.6. Analysis of the Best Network

From the input videos, the spatial features of the independent frames conveyed information about the depicted scenes and objects, whereas the temporal features of the frame sequences deal with the information of motion and movement of the objects. A 2D-CNN can learn various spatial features (e.g., edges, corners, and textures) by combining the input frame with a number of filters. The 2D-CNN is highly effective in extracting spatial features from individual frames of a video, but it is not well-suited for capturing temporal information. To accurately capture the temporal dynamics of objects in a video, a different type of neural network must be utilized. A long short-term memory (LSTM) network is a better choice for capturing temporal information. A LSTM network is a deep learning architecture based on an artificial recurrent neural network (RNN). It was specifically designed to handle sequential data, including videos, when modeling the short-range and long-range relationships of sequence features [66]. It also resolves the gradient vanishing problem of the RNN. It is usually used for time series predictions [67]. However, to apply an LSTM network for temporal feature extraction, the output of the 2D-CNN spatial feature extractor can be fed to the LSTM network as input [66]. This can be performed by utilizing the output of the last fully connected layer of the 2D-CNN as the input for the LSTM. In this fashion, the LSTM network can utilize the spatial information extracted by the CNN, together with its capacity to recall past inputs to make predictions regarding the temporal relationships in the video.

Both RNNs and LSTMs are laborious to train because they need memory-bandwidth-bound computation, which is laborious for hardware designers and eventually limits the applicability of neural networks solutions. By combining 2D-CNN and LSTM, it is possible to extract both spatial and temporal features from a videos. One of the reasons why researchers are more partial to using 2D-CNN over LSTM is the amount of training time required. The contemporary generation of well-known deep learning hardware applications mostly use Nvidia graphics cards, and they are optimized for processing 2D data with the greatest possible parallelism and speed, which 2D-CNN brings into service. Nevertheless, one of the main disadvantages of LSTM is its inability to handle temporal dependencies that are longer than a few steps. For example, when an LSTM was trained on a dataset with long-term dependencies (e.g., 100 steps), the network struggled to learn the task and generalize to new examples [68]. Furthermore, on the whole, when data are scarce or noisy, an LSTM tends to overfit the training data and suffers the loss of generalization ability [69]. As a result, it is discouraged to use an LSTM for extracting temporal features. A better solution for extracting temporal features is to employ a C3D network. For example, to take advantage of a 2D-CNN architecture, all filters and pooling kernels of 2D-CNN models can be inflated to a 3D-CNN, by equipping them with an additional temporal dimension, i.e., η×η filters become η×η×η filters. Afterwards, the weights of 2D filters can be repeated η times along the temporal dimension, to bootstrap parameters from pretrained 2D-CNN models to the 3D-CNN models [70].

We propose TSAN, which generates reweighed attention features by measuring the degree of abnormality of snippets. Explicitly, the mechanism of TSAN maximizes attention on a subset of features, while minimizing the attention on noise. To a large extent, our exceptional performance comes from the utilization of the TSAN along with the fusion of the features of I3D and the rich contextual vision-language features of CLIP.

Most of the existing approaches in Table 1 encode visual content by applying a CNN-based backbone of either C3D or I3D. Like the existing C3D or I3D based models in Table 1, our proposed C3D-TSAN and I3D-TSAN models demonstrated a performance of a comparable nature. Nevertheless, the I3D-TSAN model showed superior performance to the C3D-TSAN model on identical setups. The C3D was more suitable for spatiotemporal feature learning compared to the 2D CNN [27]. Fundamentally, the operation of 2D convolution tries to convolve an image and the 2D convolution kernel to extract the spatial features from an image, whereas the function of 3D convolution is to convolve the cube constructed by stacking several successive video frames and the 3D convolution kernel for extracting video features in the spatiotemporal dimension. More specifically, the C3D is an excellent model for applying 3D convolution kernels, which is natural for processing signals with spatiotemporal features, such as videos. Even so, its complicated structure stops it becoming deeper [71]. The I3D is an improved model based on C3D. Basically, I3D puts into practice an inflated version of the inception module architecture [30]. The fundamental features of the inception module are the employment of the incorporated effects of filters with various sizes and pooling kernels, all in one layer; as well as the manipulation of 1×1 convolutional filters, which not only assist in lessening the number of parameters but also put in place updated combinations of features to the next layers. This reveals the fact that the performance of the I3D-TSAN network is better than that of the C3D-TSAN, due to the improved architecture and more generalized features of the I3D.

On the other hand, the ViT based CLIP-TSAN model showed the best performance among the three proposed models of C3D-TSAN, I3D-TSAN, and CLIP-TSAN. Both C3D and I3D have a traditional method of convolution, where some channels may be less useful information and consume computational power [72]. Basically, both C3D and I3D were pretrained on action recognition tasks. Differently from the action recognition problem, video anomaly detection depends on discriminative representations that clearly present the events in a scene. Thus, the existing C3D and I3D backbones are not suitable due to the issue of domain gap [1]. To explain this impediment, recently, ViT-based pretrained models (e.g., CLIP, X-CLIP, VideoSwin) were leveraged [20,34,35], which proved the effectiveness of feature representation learning. For example, the ViT-based method of Joo et al. [20] outperformed all existing CNN-based methods in Table 1. Similarly, our proposed CLIP-TSAN model showed almost the same performance as the model of Joo et al. [20]. Our proposed model C3D-CLIP-TSAN demonstrated a better performance than CLIP-TSAN, due to the information fusion [4] from CNN and ViT. Nevertheless, the C3D-CLIP-TSAN model showed slightly inferior performance to I3D-CLIP-TSAN on identical grounds. This was largely due to the I3D simply having a better architecture than the C3D [22]. For instance, the I3D operates on two 3D stream inputs, whereas the C3D operates on single 3D stream input [73].

### 4.7. Ablation Study

We conducted an ablation study to investigate the effectiveness of the Mahalanobis metric for our generalized framework of CNN-ViT-TSAN. We conducted the experiments in two cases: (i) with the Mahalanobis metric and (ii) without the Mahalanobis metric but with a mean feature magnitude of snippets for identical configuration settings. Table 2 reports their performance. From Table 2, it can be observed that the maximum 5.01%, 5.18%, 4.99%, 5.25%, and 5.56% performance gains were obtained for the UMN, UCSD-Ped1, UCSD-Ped2, ShanghaiTech, and UCF-Crime datasets by applying the Mahalanobis metric. In summery, for these datasets, on average, a maximum 5% better performance could obtained empirically by employing the Mahalanobis metric (ı.e., without using the mean snippet feature magnitude).

### 4.8. Limitation of Our Model

Our WVAED models utilize extracted feature representations using CNN- and/or ViT-based pretrained feature extractors as input. As a result, the performance of our models partially depends on the pretrained feature extractors, making the calculation costly. In the testing phase, if the length of a snippet is Δ frames, then less than Δ frames video clips can be discarded or padded with the final label of the video. In this paper, we chose the former case with Δ=16 frames. Thus, less than 16 frames of video clips were ignored, which might contain useful information for performance evaluation.

## 5. Conclusions

We proposed an MIL-based generalized architecture named CNN-ViT-TSAN by applying CNN- and/or ViT-extracted features and the use of TSAN, to design a series of deep models for the WVAED problem. Our proposed TSAN mechanism minimized the attention on noise but maximized attention on a subset of features. Instead of using the mean feature magnitude, we uniquely introduced the usage of the Mahalanobis distance for the WVAED problem. At least a 5% performance gain was empirically recorded by employing the Mahalanobis distance with an identical setup as for the mean snippet feature magnitude. The information fusion between CNN and ViT was a unique contribution of this paper. Our deep models possessed a distinct degree of feature extraction ability and usability. One of our models (I3D-CLIP-TSAN) was capable of utilizing a better quality of features and confirmed a high separability between normal and abnormal snippets for VAED. The empirical results from several publicly available crowd datasets demonstrated the generalization ability and applicability of our models against the state-of-the-art approaches to the WVAED problem.

Fundamentally, our model is a natural extension of video classification based on pretrained feature extractors from CNN and ViT. ViT technology has been gaining great interest and its utilization has spread broadly in computer vision. It is assumed that ViT can better capture long-range contextual relationships in videos. We employed CLIP [26] as a ViT feature extractor, and other options including VisualBERT [31], ViLBERT [32], and data efficient CLIP [33] could be employed. Recently, the XD-Violence [10] dataset has become a common benchmark for WVAED [10,19,20]. However, we could not use the XD-Violence [10] dataset due to some nontechnical reason regarding its accessibility (e.g., not being approved by the Norwegian Data Protection Authority); however, in future, we wish to test our models with it.

## Figures and Tables

**Figure 1 sensors-23-07734-f001:**
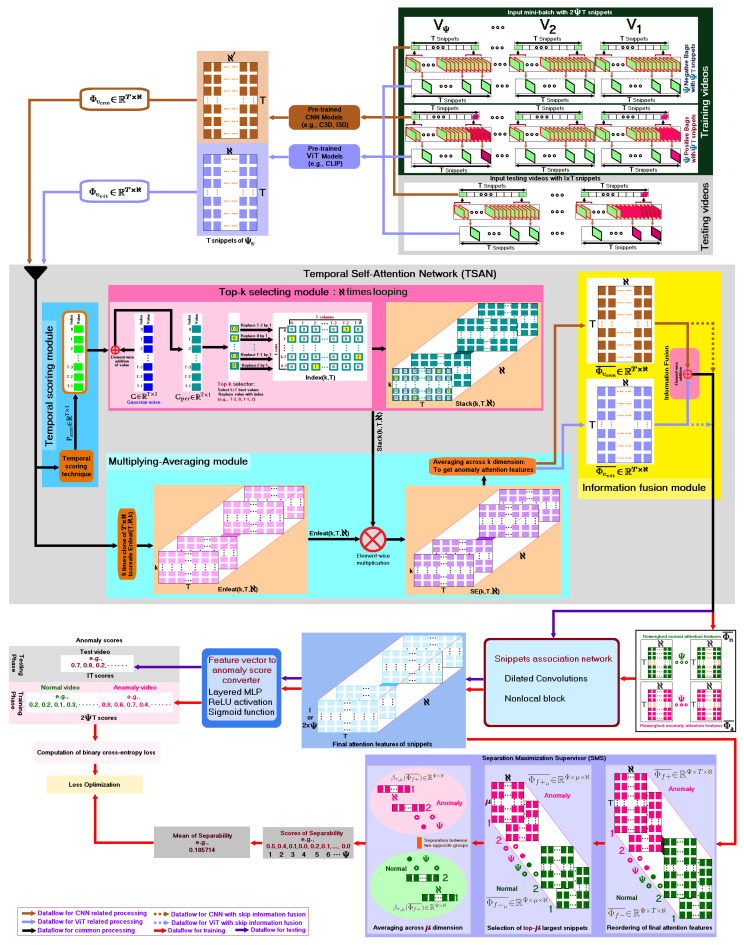
Generalized architecture of our proposed CNN-ViT-TSAN framework.

**Figure 2 sensors-23-07734-f002:**
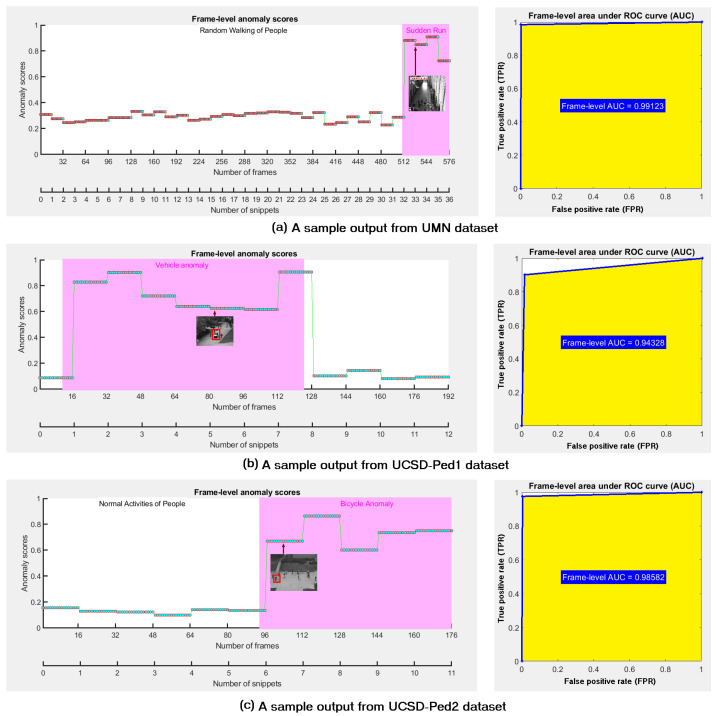
Visualization of sample testing results with various datasets. Pink regions show the manually labeled abnormal events, while the yellow regions indicate the areas below the ROC curves.

**Figure 3 sensors-23-07734-f003:**
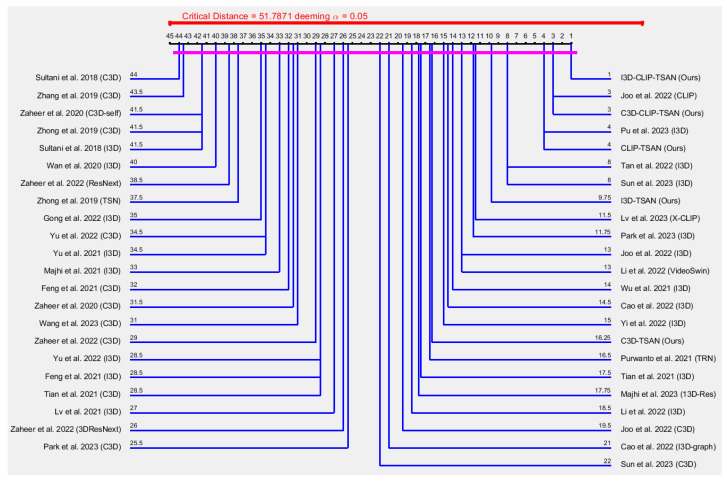
Nemenyi [64] post hoc critical distance diagram with α=0.05 using $1-AUC_{o}$ scores in Table 1 for the ShanghaiTech and UCF-Crime datasets.

**Figure 4 sensors-23-07734-f004:**
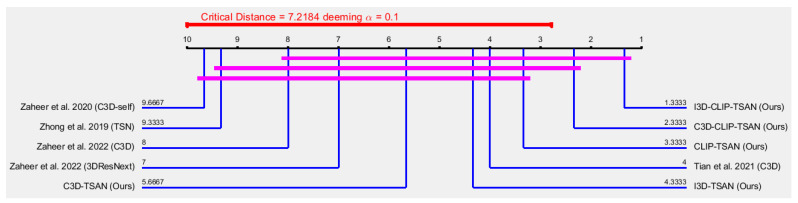
Nemenyi [64] post hoc critical distance diagram with α=0.10 using the $1-AUC_{o}$ scores in Table 1 for the UCSD-Ped2, ShanghaiTech, and UCF-Crime datasets.

**Figure 5 sensors-23-07734-f005:**
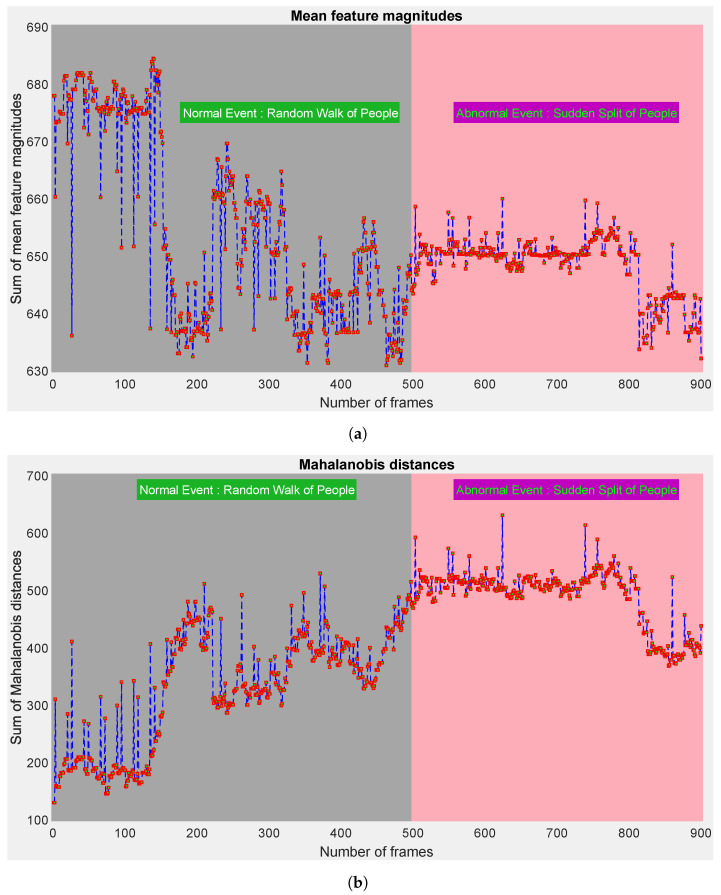
Numerical comparison of mean feature magnitudes and Mahalanobis distances. (**a**) Normal and abnormal frames cannot be distinguished using mean feature magnitudes. (**b**) Mahalanobis Distances can somewhat make difference between normal and abnormal frames.

**Table 1 sensors-23-07734-t001:** Frame-level $AUC_{o}$ score comparison of the various weakly-supervised methods and datasets. Column-wise the best score is bolded and the second best score is underlined.

Year	Weakly Supervised Model	Feature	Frame-Level Performance Scores from Different Datasets					
			**UCSD-Ped2**		**ShanghaiTech**		**UCF-Crime**	
			${\mathrm{AUC}}_{o}$	$1-{\mathrm{AUC}}_{o}$	${\mathrm{AUC}}_{o}$	$1-{\mathrm{AUC}}_{o}$	${\mathrm{AUC}}_{o}$	$1-{\mathrm{AUC}}_{o}$
2018	Sultani et al. [8]	C3D	—	—	0.8317	0.1683	0.7541	0.2459
	Sultani et al. [8]	I3D	—	—	0.8533	0.1467	0.7792	0.2208
2019	Zhong et al. [6]	C3D	—	—	0.7644	0.2356	0.8108	0.1892
	Zhong et al. [6]	TSN	0.9320	0.0680	0.8444	0.1556	0.8212	0.1788
	Zhang et al. [9]	C3D	—	—	0.8250	0.1750	0.7870	0.2130
2020	Zaheer et al. [46]	C3D-self	0.9447	0.0553	0.8416	0.1584	0.7954	0.2046
	Zaheer et al. [7]	C3D	—	—	0.8967	0.1033	0.8303	0.1697
	Wan et al. [47]	I3D	—	—	0.8538	0.1462	0.7896	0.2104
2021	Purwanto et al. [13]	TRN	—	—	0.9685	0.0315	0.8500	0.1500
	Tian et al. [19]	C3D	—	—	0.9151	0.0849	0.8328	0.1672
	Majhi et al. [48]	I3D	—	—	0.8822	0.1178	0.8267	0.1733
	Tian et al. [19]	I3D	**0.9860**	**0.0140**	0.9721	0.0279	0.8430	0.1570
	Wu et al. [49]	I3D	—	—	0.9748	0.0252	0.8489	0.1511
	Yu et al. [50]	I3D	—	—	0.8783	0.1217	0.8215	0.1785
	Lv et al. [12]	I3D	—	—	0.8530	0.1470	0.8538	0.1462
	Feng et al. [51]	C3D	—	—	0.9313	0.0687	0.8140	0.1860
	Feng et al. [51]	I3D	—	—	0.9483	0.0517	0.8230	0.1770
2022	Zaheer et al. [3]	ResNext	—	—	0.8621	0.1379	0.7984	0.7984
	Zaheer et al. [52]	C3D	0.9491	0.0509	0.9012	0.0988	0.8337	0.1663
	Zaheer et al. [52]	3DResNext	0.9579	0.0421	0.9146	0.0854	0.8416	0.1584
	Joo et al. [20]	C3D	—	—	0.9719	0.0281	0.8394	0.1606
	Joo et al. [20]	I3D	—	—	0.9798	0.0202	0.8466	0.1534
	Joo et al. [20]	CLIP	—	—	0.9832	0.0168	0.8758	0.1242
	Cao et al. [53]	I3D	—	—	0.9645	0.0355	0.8587	0.1413
	Li et al. [34]	I3D	—	—	0.9608	0.0392	0.8530	0.1470
	Cao et al. [54]	I3D-graph	—	—	0.9605	0.0395	0.8467	0.1533
	Tan et al. [55]	I3D	—	—	0.9754	0.0246	0.8671	0.1329
	Li et al. [34]	VideoSwin	—	—	0.9732	0.0268	0.8562	0.1438
	Yi et al. [56]	I3D	—	—	0.9765	0.0235	0.8429	0.1571
	Yu et al. [57]	C3D	—	—	0.8835	0.1165	0.8208	0.1792
	Yu et al. [57]	I3D	—	—	0.8991	0.1009	0.8375	0.1625
	Gong et al. [58]	I3D	—	—	0.9010	0.0990	0.8100	0.1900
2023	Majhi et al. [59]	13D-Res	—	—	0.9622	0.0378	0.8530	0.1470
	Park et al. [60]	C3D	—	—	0.9602	0.0398	0.8343	0.1657
	Park et al. [60]	I3D	—	—	0.9743	0.0257	0.8563	0.1437
	Pu et al. [61]	I3D	—	—	0.9814	0.0186	0.8676	0.1324
	Lv et al. [35]	X-CLIP	—	—	0.9678	0.0322	0.8675	0.1325
	Sun et al. [62]	C3D	—	—	0.9656	0.0344	0.8347	0.1653
	Sun et al. [62]	I3D	—	—	0.9792	0.0208	0.8588	0.1412
	Wang et al. [63]	C3D	—	—	0.9401	0.0599	0.8148	0.1852
	C3D-TSAN (Ours)	C3D	0.9675	0.0325	0.9608	0.0392	0.8578	0.1422
	I3D-TSAN (Ours)	I3D	0.9758	0.0242	0.9743	0.0257	0.8650	0.1350
	CLIP-TSAN (Ours)	CLIP	0.9811	0.0189	0.9806	0.0194	0.8763	0.1237
	C3D-CLIP-TSAN (Ours)	C3D+CLIP	0.9824	0.0176	0.9813	0.0187	0.8802	0.1198
	I3D-CLIP-TSAN (Ours)	I3D+CLIP	0.9839	0.0161	**0.9866**	**0.0134**	**0.8897**	**0.1103**

**Table 2 sensors-23-07734-t002:** Ablation study of Mahalanobis metric on various datasets. Column-wise the best score is bolded and the second best score is underlined.

Feature	Mahalanobis Metric Included?	Frame-Level Performance Scores from Different Datasets									
		**UMN**		**UCSD-Ped1**		**UCSD-Ped2**		**ShanghaiTech**		**UCF-Crime**	
		${\mathrm{AUC}}_{o}$	**Gain**	${\mathrm{AUC}}_{o}$	**Gain**	${\mathrm{AUC}}_{o}$	**Gain**	${\mathrm{AUC}}_{o}$	**Gain**	${\mathrm{AUC}}_{o}$	**Gain**
C3D	No	0.9136	1.00	0.8553	1.00	0.9214	1.00	0.9129	1.00	0.8262	1.00
	Yes	0.9517	4.17%	0.8996	**5.18%**	0.9675	**4.99%**	0.9608	**5.25%**	0.8578	3.82%
I3D	No	0.9362	1.00	0.8903	1.00	0.9489	1.00	0.9359	1.00	0.8401	1.00
	Yes	0.9644	3.01%	0.9085	2.04%	0.9758	2.83%	0.9743	4.09%	0.8650	2.96%
CLIP	No	0.9417	1.00	0.9063	1.00	0.9597	1.00	0.9391	1.00	0.8346	1.00
	Yes	0.9731	3.33%	0.9274	2.33%	0.9811	2.23%	0.9806	4.42%	0.8763	4.99%
C3D+CLIP	No	0.9405	1.00	0.8871	1.00	0.9396	1.00	0.9422	1.00	0.8348	1.00
	Yes	0.9876	**5.01%**	0.9315	5.01%	0.9824	4.56%	0.9813	4.15%	0.8812	**5.56%**
I3D+CLIP	No	0.9461	1.00	0.8943	1.00	0.9448	1.00	0.9400	1.00	0.8462	1.00
	Yes	**0.9903**	4.67%	**0.9402**	5.13%	**0.9839**	4.14%	**0.9866**	4.96%	**0.8897**	5.14%

## Data Availability

The datasets used in this study are openly available and downloadable from http://mha.cs.umn.edu/proj_events.shtml#crowd, http://www.svcl.ucsd.edu/projects/anomaly/dataset.htm, www.cse.cuhk.edu.hk/leojia/projects/detectabnormal/dataset.html, https://svip-lab.github.io/dataset/campus_dataset.html, and https://webpages.charlotte.edu/cchen62/dataset.html, accessed on 28 March 2023. Those datasets, expect http://mha.cs.umn.edu/proj_events.shtml#crowd, were approved by the Sikt (Norwegian Agency for Shared Services in Education and Research) with the reference number of 720663.

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
