# Peer review of "CNN-ViT Supported Weakly-Supervised Video Segment Level Anomaly Detection"

_sensors, 2023, doi:10.3390/s23187734_

Round 1
Reviewer 1 Report
The manuscript proposed weakly-supervised video segment-level anomaly detection based on the extracted visual features. Here, the visual features are extracted using CNN and ViT. The overall illustration is logical and clear. Several minor issues:
1. 3.1.3 Feature length normalization for training can be put in 3.3. training part
2. How you use C3D (the frame length) can add more details here.
3. Lin 212-213 how features from CNn and ViT is separately converted into a probability score vector Pscore is not clear. Suggest adding more description.
4. 4.8. Future work can be put in Section 5
5. added some discussion as a new section 4.8, why not use LSTM to extract temporal features? added some discussion here?
6. in the testing phase, if the video length is not the same, your method will still work or not? Like to know how to deal with virant-length videos?
Reviewer 2 Report
This paper presents a novel approach, called CNN-ViT-TSAN, for weakly supervised video anomaly event detection (WVAED). The proposed method leverages pre-trained feature extractors (C3D, I3D, and CLIP) to extract effective representations. It incorporates both long-range and short-range temporal dependencies using a TSAN. Experimental results on popular crowd datasets prove the effectiveness of the CNN-ViT-TSAN approach in WVAED. Overall, this paper is well-structured and of great value. Nevertheless, there are a few issues that should be acknowledged and addressed.
1. In the Introduction section, it is crucial for the authors to provide a more specific and targeted explanation of the problem they are addressing. They should clearly articulate the necessity of Weakly-Supervised Video Segment Level Anomaly Detection. What are the limitations or challenges with existing methods that make weakly-supervised approaches necessary?
2. When describing the proposed method, it is crucial to emphasize the key innovations of your work and how they distinguish it from related approaches. Highlight the unique contributions and advancements that your method brings to the field.
3. In application scenarios of anomaly detection, efficiency is an important factor to consider. Therefore, I suggest conducting a comprehensive time-consuming comparison to evaluate the efficiency of the proposed method. This comparison can demonstrate the computational performance and runtime of the method in comparison to existing approaches. By including an efficiency analysis, you provide valuable insights for practical implementation and adoption of the proposed method in real-world scenarios.
4. The experimental results should be thoroughly discussed to highlight the specific contribution of the proposed framework. Emphasize how the proposed method outperforms existing methods or addresses limitations in the field. Discuss any notable findings, trends, or patterns observed in the results.
5. The Conclusion section can be improved by presenting the principles demonstrated by the results more comprehensively. Clearly articulate the key findings and their significance for the research field. Discuss the theoretical implications of your work, highlighting the unique contributions made by this article.
Reviewer 3 Report
Authors designed a MIL-based generalized framework of CNN-ViT-TSAN to specify a series of models for WVAED problem. The structure of CNN-ViT-TSAN was illustrated. Experimental were carried out on UMN, UCSD-Ped1, UCSD-Ped2, ShanghaiTech, and UCF-Crime datasets. Detailed results were presented. There are several points should be improved.
1. Innovations are expected to be enhanced. Five deep models were obtained by combining the CNN and ViT based pre-trained models. The experiments showed I3D-CLIP-TSAN has the best results. However, the reasons why such network is best were not analyzed. Similarly, in Section 3.2, although Temporal self-attention network was introduced, further analyses were neglected.
2. In Section 4.1, the completed introduction of datasets seems not necessary.
3. What is the definition of AUC in Section 4? In Section 4.3 AUC scores obtained by I3D-CLIP-TSAN when processing the videos of UCSD-Ped2, ShanghaiTech, and UCF-Crime datasets are 0.986, 0.989, and 0.912 respectively. However we can not find the corresponding results in Table.1. Please explain it.
4. In Figure 2, the two horizontal axes indicating the number of frames and the number of snippets in every subfigure are confusing.
Round 2
Reviewer 3 Report
What is the definition of AUC in Section 4? Authors are recommended to present its calculation formula.
